# Gemini and Bicephalous Surfactants: A Review on Their Synthesis, Micelle Formation, and Uses

**DOI:** 10.3390/ijms23031798

**Published:** 2022-02-04

**Authors:** Lluvia Guerrero-Hernández, Héctor Iván Meléndez-Ortiz, Gladis Y. Cortez-Mazatan, Sandra Vaillant-Sánchez, René D. Peralta-Rodríguez

**Affiliations:** 1Centro de Investigación en Química Aplicada, Blvd. Enrique Reyna No. 140, Col. San José de los Cerritos, Saltillo 25294, Mexico; lluvia.guerrero.d20@ciqa.edu.mx (L.G.-H.); gladis.cortez@ciqa.edu.mx (G.Y.C.-M.); vaillant39458@gmail.com (S.V.-S.); 2CONACyT—Centro de Investigación en Química Aplicada, Blvd. Enrique Reyna No. 140, Col. San José de los Cerritos, Saltillo 25294, Mexico

**Keywords:** micelles, amphiphiles, gemini surfactants, bicephalous surfactants, polymer, drug nanocarriers

## Abstract

The use of surfactants in polymerization reactions is particularly important, mainly in emulsion polymerizations. Further, micelles from biocompatible surfactants find use in pharmaceutical dosage forms. This paper reviews recent developments in the synthesis of novel gemini and bicephalous surfactants, micelle formation, and their applications in polymer and nanoparticle synthesis, oil recovery, catalysis, corrosion, protein binding, and biomedical area, particularly in drug delivery.

## 1. Introduction

According to IUPAC definition, a micelle is a “Particle of colloidal dimensions that exists in equilibrium with the molecules or ions in solution from which it is formed” [1,2]. Further, micelles are formed by spontaneous aggregation (self-assembly, supramolecular assemblies) of amphiphilic molecules that contain a hydrophilic/polar region (head) and a hydrophobic/nonpolar region (tail). Besides forming micelles, amphiphiles can self-assemble in different structures, such as vesicles, nanotubes, nanofibers, and lamellae [3]. In this contribution, the formation of micelles from two families of non-conventional amphiphiles, gemini and bicephalous surfactants, is reviewed, and their applications in polymer synthesis in dispersed media, catalysis, protein binding, and as drug carriers are presented, as well as future developments.

## 2. Surfactants

Surfactants are amphiphilic molecules that have hydrophobic (head) and hydrophilic (tail) components, allowing their solubility in both organic solvents and water. At the air–water interface, the hydrophobic tail is in the air and the hydrophilic head in the water, causing a decrease of both, in the surface tension, which is defined as the force of attraction between the molecules at the air–water interface, and in the interfacial tension between two liquids [4,5]. Usually, the hydrophobic tail is a relatively long hydrocarbon, fluorocarbon, or siloxane chain, while the hydrophilic part could be an ion (cation or anion).

Surfactants are key compounds in many industrial processes (lubricants, foaming agents, wetting agents, solubilizers, corrosion inhibitors, antistatic agents, and viscosity modifiers) and a variety of useful products (disinfectants, emulsifiers, dispersants, detergents, and soaps) have been vigorously developed in terms of functional variety and structural diversity in the last few years [6,7,8].

### 2.1. Classification of Surfactants

These compounds may be classified based on the chemical nature of their polar head [9]. If the head group has no charge, the surfactant is called non-ionic. They can be classified as anionic, cationic, non-ionic, and zwitterionic (Figure 1)

#### 2.1.1. Anionic Surfactants

In this type of surfactants, the hydrophilic group has a negative charge on the polar head, such as carboxylate (RCOO^−^), sulphonate (RSO^3−^), or sulphate (RO-SO^3−^) [10]. When these surfactants dissolve in water, negatively charged particles (anions) are created. Anionic surfactants are widely used for industrial as well as household cleaning and for pesticide formulations;for example, potassium laurate, sodium lauryl sulphate, sodium decyl sulfate, sodium N-lauroyl-N-methyltaurate, sodium tetradecyl sulphate, sodium stearate, α-olefin sulfonate, etc. (Figure 2).

#### 2.1.2. Cationic Surfactants

These surfactants possess a positive charge on the polar head, which may be either permanent or only exist in a range of pH values. The cationic surfactants can dissociate in water with the formation of surface-active cations [11]. One of the advantages of the cationic surfactants is the diversity of their head groups, which permits the chemical modification and introduction of desirable moieties. In addition, these surfactants have compatibility with both non-ionic and amphoteric surfactants and incompatibility with the ionic ones. Usually, cationic surfactants are found in fabric softener formulations, antistatic agents, particle dispersants, corrosion inhibitors, and emulsifiers [12,13]. Cationic surfactants also have found important application in pharmacy and biomedicine as drug nanocarriers [14,15,16]; for example, cetyltrimethylammonium bromide, cetylpyridinium chloride, and alkyldimethyl amine oxides (Figure 3).

#### 2.1.3. Non-Ionic Surfactants

The non-ionic surfactants have polar head groups that are not electrically charged. In general, the solubility of non-ionic surfactants in water is not good compared with the solubility of ionic surfactants; however, they do not change the pH of the solution. Usually, these surfactants show a better biocompatibility than ionic ones, making them suitable for biomedical applications [17,18,19,20]. The non-ionic surfactants are also the most used in the food industry [21,22,23]. Typical non-ionic surfactants are Tween 20, Tween 80, Triton X-100, Brij-35, and alkylethers of poly(ethylene glycol) and poly(propylene glycol) (Figure 4).

#### 2.1.4. Amphoteric or Zwitterionic Surfactants

This type of surfactant contains both positive and negative polar heads. They are surfactants with a zero net charge, so the surfactant molecule is essentially neutral [24,25]. Usually, the positive head is either an amine or a quaternary ammonium cation, whereas the anionic part is mostly a carboxylic, sulfuric, or phosphoric acid functional group [26,27]. Some examples of amphoteric surfactants are the carboxilates RCOO- with quaternary amine (R4N^+^), phospholipids, betaines, and sulfobetaines (Figure 5).

### 2.2. Hydrophile–Lipophile Balance (HLB)

The appropriate determination of the hydrophilic–lipophilic nature of surfactants plays an essential instrumental role in leading the way for the formulation of emulsions and microemulsions. The hydrophile–lipophile balance (HLB) approach has been used to measure the degree of hydrophilicity (tendency to solubilize in water) or lipophilicity (tendency to solubilize in oil) of surfactants. The HLB numbers are determined by calculating values for the hydrophilic and lipophilic regions of the molecule, as described by Griffin [28,29,30]. The range of values for this parameter goes from 0 to 20 [31]. In this sense, surfactants with low HLB values tend to be more lipophilic while surfactants with high HLB values are more soluble in water. The HLB value also can be used to predict the potential application for a given surfactant; e.g., a value in the range of 0–3 indicates an antifoaming agent while a range of 13–15 is typical of detergents [32]. Surfactant formulation development based on the HLB approach has worked well for ethoxylated, non-ionic surfactants but not so for the ionic ones.

## 3. Micelles

One of the most important properties of surfactants is their capacity to self-assembly to create nanometer size structures. When the concentration of surfactant molecules exceeds the limit of their solubility, these molecules become organized themselves into nanomeric structures called micelles [33,34]. 

Micelles are commonly defined as core-shell surfactant-based systems dispersed in a bulk phase. Surfactants can spontaneously create these nanometric systems in either an aqueous or oily phase. The micelles formed in aqueous solution are called conventional (or normal) while those formed in an oily bulk phase are called reverse (or inverse). In the conventional micelles, the shell is bordered by the hydrophilic region of the surfactant molecules, while the hydrophobic one forms the core (Figure 6). Micelles can be spherical, cylindrical, or organized in multi-layered flat sheets. In addition, the morphology of the micelles can be tuned by varying some parameters, including the size and type of the hydrophobic tail of the surfactant, the nature and size of the polar head, as well as the concentration, temperature, pH, etc. [35,36,37].

Micellar solutions are used in some important applications, such as tertiary oil recovery [38,39,40], catalysis [41,42,43], food and cosmetics formulations [44,45,46], pharmaceutical drug delivery systems [47,48,49,50], polymer synthesis [51], etc. Pharmaceutical application of micelles is of considerable interest regarding its importance in biological systems in which therapeutic efficiency is crucial. Micelles can be employed to solubilize drugs, and thus increase their bioavailability. This depends upon the interaction between the drug and micellar core as well as the stability of the system being formed [52,53]. Likewise, the application of micelles in polymer synthesis is of utmost importance: a wealth of useful polymers is synthesized in monomer swollen micelles; e.g., emulsion polymers via the micellar nucleation mechanism [51].

### 3.1. Critical Micellar Concentration (CMC)

The CMC is defined as the concentration of dissolved surfactant molecules above which aggregates, called micelles, are spontaneously formed (self-assembly) [54,55]. At the CMC, small spherical micelles are typically formed while at larger concentrations, they may grow to worm- or vesicle-like micelles. At concentrations above the CMC, the micelles are in dynamic equilibrium with free molecules but are thermodynamically stable and tend to resist disassembly [56]. In addition, upon reaching the CMC, any further addition of surfactants will just increase the number of micelles. On the contrary, below the CMC, the micelles are dissociated at a rate that depends mainly on the nature of the surfactant and the degree of interaction between the surfactant molecules. For a given system, micellization occurs over a narrow concentration range. Besides concentration, temperature also influences micelle formation, and the temperature corresponding to the initiation of micelle formation is designated as the Krafft point [4,5].

Usually, low-molecular weight surfactants exhibit higher CMC values than the high-molecular weight and block copolymer surfactants, which show a greater resistance to dissociation upon dilution [56]. The micelles formed by block copolymer surfactants generally present a core-shell morphology wherein the hydrophobic segments form the core within a size range of 10–100 nm.

### 3.2. Aggregation Number

The aggregation number is defined as the average number of surfactant molecules constituting a micelle once the CMC has been reached [57,58]; it also gives information about the micelle size and shape, which are important in determining their stability and applications. This number can be affected by temperature, pH, type of surfactant, the addition of either electrolytes or organic compounds, etc. Several methods have been reported for the determination of the micellar aggregation number, including light-scattering [59], fluorescence [35,60] transmission electron microscopy [61,62], isothermal titration calorimetry [63,64], small-angle neutron scattering (SANS) [65,66,67], among others.

## 4. Gemini Surfactants

### 4.1. Definition

Gemini surfactants are dimeric structures, composed of two hydrophobic chains and two hydrophilic heads, linked by a spacer at or near the head groups (Figure 7). They present lower CMC, better efficiency to form micelles, and solubilization capacity comparedto their conventional (monomeric) counterparts [68,69,70]. They can also reduce the surface tension of water and the oil–water interfacial tension from 10 to 100 times. This behavior depends mainly on the nature of their components (heads, hydrophobic chains and spacer); thus, their synthesis is focused mainly on varying the type and length of these components.

### 4.2. Structure

Gemini surfactants have attracted interest among the scientific community in various applications due to their very low CMC, greater solubilization power, as well as better wetting and foaming properties compared to single-chain surfactants [71]. Gemini surfactants have a polymorphic phase behavior and a great variety of self-assembled structures forming aggregates that can be observed as micelles, bilayers, and vesicles, depending on the head groups, the size of the hydrophobic tails, and the nature of the spacer [72]. Figure 8 shows the structure of normal micelles obtained from conventional and gemini surfactants.

### 4.3. Type of Gemini Surfactants

Gemini surfactants are classified by their physicochemical characteristics, groups present in the hydrophobic tails, and spacers. Regarding rigidity, the spacers in the chemical structure of a gemini surfactant can be classified into two subcategories, flexible (methylenes) and rigid (stilbene) (Figure 9a,b, respectively). Spacers also can be classified according to their length into short (Figure 9c) or long (Figure 9d). It is worth mentioning that the length of the spacer influences the geometry of the micelles. The presence of short spacers increases the repulsion between the head groups, resulting in micelles with a fiber-like structure, even at low concentrations of surfactant (Figure 10a). On the contrary, when the spacers are long, the micelles have elliptical geometries (Figure 10b). The transition from spherical micelles (Figure 10c) (4–8 carbon atoms in the spacer) to elliptical micelles occurs when repulsion between the groups of the polar heads decreases [73].

On the other hand, the groups present in the spacer can be classified into polar (Figure 9e) or nonpolar (aliphatic and aromatic groups). Furthermore, the polar head can be positive, negative, zwitterionic, or non-ionic (Figure 9f–i). Finally, gemini dissymmetric (heterogeminis) surfactants contain two groups of non-identical polar heads (or identical) and different (or identical) lengths of alkyl tails, so they can also be classified into gemini surfactants of different head or hydrophobic tails and gemini surfactants of identical head and hydrophobic tails (Figure 9j–l) [74].

The surface activity of heterogeneous surfactants is highly dependent on the degree of asymmetry. For pyrene-based asymmetric gemini surfactants synthesized in five-step reactions, the Krafft temperature increases as the alkyl chain length increases. Similarly, the CMC values are much lower than their symmetric counterparts [75].

### 4.4. Synthesis Pathways

There are three main routes to synthesize symmetric gemini surfactants (Figure 11): (**a**) reaction of long chain tertiary amines with dihalogenated substrates as organic dibromides or dichlorides; (**b**) reaction of N,N,N′,N′-tetramethylpolymethylene diamines with alkyl halides; and (**c**) reaction of long chain tertiary amines with a haloalkylene oxide substrate.

The yield of the synthesis of gemini surfactants depends mainly on the reactivity of the dihalogenoalkanes and the polarity and protic character of the solvent [76]. The best results have been achieved in aprotic and polar solvents. Some of these reactions can also be carried out without a solvent under mild conditions with very high yields [77]. Amino acid-based gemini surfactants are synthesized by condensation reactions at the amino group or the carboxyl group of the amino acid [78]. There are many studies on the synthesis and biological evaluation of gemini surfactants based on amino acids derived from arginine [79]. Some gemini surfactants have also been obtained from lysine, glycine, and cysteine [80,81]. Wang et al. synthesized a sugar-based gemini surfactant with a N, N′-acetylethylenediamine spacer (N,N′ (N-dodecyl-2-D-glucosaminyl acetyl) ethylenediamine and D-(+)-glucono-1,5-lactone as the starting material, in three steps. The CMC value (10^−5^ mol·L^−1^) determined by surface tension indicates a higher surface activity than the corresponding monomeric sugar-based surfactants [82]. With the aim of applying the surfactants in the oilfield, Hussain et al. [83] synthesized quaternary ammonium gemini surfactants with a different length of the spacer group (C8, C10, and C12), by solvent-free amidation of glycolic acid ethoxylate lauryl ether with 3-(dimethylamino)-1-propylamine. Similarly, Zhou et al. synthesized gemini surfactants in three steps using triethylene tetramine, fatty-acid methyl esters, ethyl chloride, N, N′-dimethyl ethylenediamine, and 3-chloro-2-hydroxypropane sulfonic acid sodium as the main raw materials to be applied in oilfields [84].

Thermodynamic and surface parameters are often evaluated for gemini surfactants. The effect of variations in the hydrophobic chain length of the gemini imidazolium surfactants on thermodynamic and surface parameters was studied by Ren et al. [85] (Figure 12). The results indicated that the micellization process could be both enthalpy and entropy driven, and that the increase in alkyl chain length causes the decreases in CMC and aggregation number.

On the other hand, the synthesis and characterization of the anionic sulfonate gemini surfactants (Figure 13) with different hydrophobic chain length shows that this kind of surfactant presentsa lower density, viscosity, and CMC than sodium dodecylbenzene sulfonate (SDBS), a monomeric surfactant with twelve carbon atoms in the hydrophobic chain [86].

To improve the biodegradability of the cationic gemini surfactants, biodegradable moieties such as ester and amide groups have been used (Figure 14). It has been found that gemini surfactants are pH-responsive in alkaline conditions due to the ester group between the cationic head groups. The cationic gemini surfactants with an ester group in the spacer are more biodegradable than those with the ester bond in the tail [87]. 

### 4.5. Micelles Formation

Gemini surfactants can produce aggregates such as micelles, bilayers, vesicles, and other structures with different additives [88]. Several authors have carried out recent studies related to the formation of aggregates from gemini surfactants due to the benefits of these surfactants compared to those with a single hydrophobic chain.

A study of the interaction of the drug amitriptyline hydrochloride and the gemini surfactant ethane-1,2-diyl bis (N,N-dimethyl-N-tetradecylammonium acetoxy) (14-E2-14) in three aqueous media showed the high ability of gemini surfactants to form spherical micelles in aqueous systems [89].

Yang et al. studied the properties of different gemini surfactants synthesized with different sizes of hydrophobic chains [90]. During the analysis, they found that the size of the aggregates formed by the surfactants increased when the surfactant concentration was raised, reaching sizes from 200 to 400 nm. In the case of studies using the transmission electron microscopy (TEM) technique, surfactants with hydrophobic chains of 12, 16, and 18 carbon atoms formed spherical groups of hundreds of nanometers in solution with a tendency to form spherical aggregates.

In 2017, Feng et al. synthesized gemini alkyl glucoside surfactants to develop vesicles using (+)—catechin (C) and (−)—epigallocatechin (EGC) laureate, finding that the thermal stability of C or EGC was improved due to the encapsulation in more ordered structures. In addition, the incorporation of these drugs at low concentrations strengthened the bilayer formed [91].

In 2018, Gan et al. reported the formation of vesicles and micelles from gemini surfactants based on glucono-δ-lactone, which depended on the length of the hydrocarbon chain as well as the surfactant concentration [92].

In addition, there are studies on the influence of some parameters, such as the concentration, pH, temperature, and the presence of salts, on the morphology of aggregates formed by cationic gemini surfactants. These studies have shown a change from micelles to vesicles and vice versa by varying either the pH or temperature. Furthermore, the presence of salts may cause a transition from vesicles to micelles (Figure 15) [93].

More recently, Asadov et al. synthesized and characterized the cationic gemini surfactant N,N′-bis(alkyl)-N,N′-bis (2-hydroxypropyl) ethylene diammonium dibromide with chain lengths of 9, 12, and 14 carbon atoms [94]. They found that the aggregate diameters decreased when temperature was increased. In another work, Rajput et al. studied the effect of the addition of diclofenac sodium to gemini surfactant micellar aggregates, reporting a transition from micelles to vesicles as a result of an increase in the drug:gemini surfactant molar ratio. They claimed that the stability of vesicles at the human body temperature also makes them candidates for use in drug release [95].

### 4.6. Applications

Gemini surfactants have found application in medicine, physics, optics, and electronics. Polymerizable gemini anionic surfactants also have been synthesized to improve its interfacial properties [96]. These surfactants have been used as a template for the synthesis of nanoparticles. Tiwari et al. described the preparation and characterization of gold, silver, and gold-silver alloy nanoparticles using gemini surfactants as stabilizers of the nanoparticles around metal surfaces [97]. In addition, gemini surfactants have been used to obtain supramolecular solvents (SUPRAS), which are nanostructured liquids formed by aggregates of surfactants obtained through a self-assembly process. This type of solvent is assigned mainly to microextraction methods with applications in the cosmetic industry [98]. On the other hand, the formation of a spatial network of well-dispersed molecules is very important for biomedical and optoelectronic applications and these surfactants have been effective to form a three-dimensional network with supramolecular micellar hybridization [99]. Furthermore, these surfactants have been used as stabilizers in enhanced oil recovery [100]. For applications in this field, sulfonates gemini surfactants were shown to reduce the oil–water interfacial tension to ultralow values, around 10^−3^ mN/m, with surfactant concentrations less than 0.5 wt % [101]. Another important parameter for applications in oilfields, is thermal stability. In this sense, Hussain et al. studied the thermal degradation of three cationic poly(ethylene oxide) gemini surfactants containing flexible and rigid spacers. The thermal gravimetric analysis showed a degradation temperature higher than that observed in an oilfield (90 °C) [102].

In the polymer area, gemini surfactants play an important role in the synthesis of hybrid systems based on surfactant-polymer materials that have different applications. Hussain et al. investigated the properties of a surfactant-polymer hybrid material as candidate for carbonate reservoir at high temperatures [103]. They studied how the spacer length of the surfactant affects the rheological properties of the surfactant-polymer solutions. Furthermore, nanoemulsions stabilized by a gemini surfactant (14-6-14 GS) have been reported [104]. In the polymerization, Dreja and Thieke [105] reported the polymerization of styrene by free radicals at 25 °C in oil-in-water microemulsions stabilized by a series of cationic dimeric (gemini) surfactants and initiated by ^60^Co-γ-radiation. The resulting polymeric dispersions contained spherical latex particles (30–60 nm average diameter) and their size could be controlled by the monomer/surfactant ratio as well as by the surfactant spacer length. The polymer weight average molecular weight varied from 0.164 to 1.400 × 10^6^ Da and depended on the spacer length and crosslinking. In a more recent study, Wang et al. [106] synthetized six quaternary ammonium salts from cardanol, a renewable resource, that can perform as gemini reactive surfactants. The surfactants, with a spacer consisting of a saturated aliphatic hydrocarbon chain, had a CMC of ≤0.2 mmol·L^−1^. A photo-active gemini surfactant with CMC = 0.05 mmol·L^−1^ was the stabilizer of a methyl methacrylate (MMA) emulsion, which was successfully polymerized using 2,2′-azobisisobutyronitrile as the initiator. Additionally, the gemini surfactant containing benzyl bromide was used as initiator and emulsifier during the atom transfer radical polymerization. The polymer obtained contained a cardanol-end unit and had an Mn = 45.1 kDa.

Regarding the use of gemini surfactants in the biomedical area, we found the research of Cardoso et al., who studied the effectiveness of complexes of serine-derived gemini surfactants and DNA in mitochondrial expression [107]. For their part, Faustino et al. reported the synthesis of gemini anionic surfactants from L-cysteine, D-cysteine, DL-cysteine, and their monomeric counterparts (Figure 16a,b), as well as the study of their properties in solution at physiological pH. In this work, gemini surfactants showed low CMC values and higher efficiency than their monomeric counterparts. Furthermore, surfactants were found to interact with bile acids, membrane phospholipids, oligosaccharides, and bovine serum albumin protein [81]. Furthermore, it has been reported that the solubilization of the drug amphotericin B (AmB) in micelles formed with an anionic gemini surfactant (derived from the amino acid cysteine) prevents self-aggregation of the drug, which makes it less toxic during administration (Figure 16c). In addition, the use of gemini surfactants avoid the use of organic solvents, often used in the preparation of other drug carriers such as polymeric micelles, liposomes, and nanoparticles [108].

Specifically, in drug delivery, Cruz et al. used cationic gemini surfactants to deliver RNA for gioblastoma treatment [109], while Michel et al. developed a cationic gemini surfactant modified with β-cyclodextrin to improve the biological and physicochemical behavior of the drug mephalan. [110]. There are some reports on the use of amino acid-derived gemini surfactants for drug delivery; in this regard, lysine-derived surfactants have been used to form niosomes as delivery systems for the parenteral administration of the anticancer drug methotrexate [111]. Srivastava et al. developed gemini surfactant vesicles for encapsulation and release of the anticancer drug doxorubicin. They found that vesicles reduce the toxicity and showed better therapeutic effects at high drug concentrations [112]. Recently, Choi et al. synthesized disulfide-bridged gemini surfactants and their micellar properties were analyzed in the release of drugs for reactive oxygen species. The self-assembled surfactants as stable micellar aggregates were subjected to a reductive environment that caused destabilization of the micelles, suggesting that this response of the micelles could be used in the release of anticancer drugs [113].

On the other hand, one of the properties of gemini surfactants that allows their uses in medicine is their antimicrobial activity, for example, against Gram-positive bacteria such as *Bacillis subtilis* and *Staphylococcus aureus* [114]. This reason makes them good capping agents for metal nanoparticles synthesis with unique and strengthened biocidal properties [67].

Cationic gemini surfactants have also found application as corrosive inhibitors (Figure 17) [115] and in the area of environmental protection, for example, in soil remediation to remove hydrophobic organic pollutants, heavy metals, and radionuclides from the soil [116].

Otherwise, the study of the interactions between proteins and surfactants is very important due to the numerous technical applications in the fields of pharmaceuticals, cosmetics, paints, coatings, etc. [117,118,119,120]. Surfactants can cause the protein conformational changes via electrostatic and hydrophobic interactions, leading to the protein folding or unfolding depending on the concentrations of surfactants and proteins [121,122,123]

Recently, gemini surfactants were shown to be more efficient to interact with proteins by comparing them with single-chain surfactants [124,125,126,127]. Zhou et al. studied the effect of the structure of cationic surfactants on the conformation of bovine serum albumin (BSA) with a series of imidazolium gemini surfactants. The results showed that the gemini surfactant with either a shorter spacer or longer chain has a larger effect on BSA unfolding, and that the interactions of BSA with imidazolium gemini surfactants are stronger than those for single quaternary ammonium surfactants [128]. For their part, Branco et al. studied the interaction between a cationic amino acid-based gemini surfactant derived from cysteine and BSA under physiological conditions [129]. 

Luo et al. focused on the investigations of the interactions between single-chain or gemini quaternary ammonium surfactants with hemoglobin.They observed that the interactions between the surfactants and hemoglobin were mainly caused by both electrostatic and hydrophobic interactions, and the hydrophobic chain length and linking group length of the surfactants had a significant influence on tuning the conformations of hemoglobin [130]. For their part, Amiri et al. reported the interactions of gemini surfactants with ribonuclease Sa, and the results indicated that the tune of protein conformations is changed with the structure of surfactants and proteins [131]. More recently, Aslam et al. reported the preparation of pyridinium-based gemini surfactants and the study of interaction with BSA. They found a strong interaction between the gemini surfactants and protein due to the decrease of the CMC of surfactant as the BSA concentration was increased [132].

Micellar catalysis is a process that consists of the accumulation of a catalyst in the internal part of a micelle [133]. The micellar catalysis was shown to improve the reaction rate between the oil–water interphase and selectivity of the target molecules in organic reactions, such as electrophilic and nucleophilic substitution, hydrolysis, etc. [134,135].

Micellar catalysis using gemini surfactants was shown to have high catalytic efficiency and accelerates processes reducing the generation of secondary reactions [136,137]. Bunton et al. proposed for the first time the use of gemini surfactants in micellar catalysis [138]. The gemini surfactant synthesized by this group showed a better catalytic efficiency than CTAB in nucleophilic substitutions reactions. Since then, morestudies have been reported [139,140,141,142]. Micellar catalysis using gemini surfactants has been applied in reactions of ester hydrolysis [143], chloromethylation [144], and nucleophilic and electrophilic substitutions [137]. Furthermore, the catalytic properties of these surfactants have favored the development of aqueous micellar catalytic processes, where the substitution of organic solvents for water is achieved, contributing to the development of more sustainable and environmentally friendly processes [145].

## 5. Bicephalous Surfactants

### 5.1. Definition and Structure

Conventional surfactants are amphiphilic organic compounds that have a hydrophilic head and hydrophobic tail whose main functions are to reduce the interfacial tension in a colloidal system, forming an interface between the two immiscible phases [146]. They are also responsible for promoting the formation of micelles [147]. The properties of each surfactant can be modified by various factors, such as pressure, temperature, and the molecular structure of the compound [148]. 

In recent years, novel surfactants have been reported with different structural arrangements by comparing with the conventional ones. These new structures consist of two hydrophilic heads, a hydrophobic tail, and a spacer that prevents repulsion, which have been called “bicephalous” surfactants [149]. There are two type of bicephalous surfactants: dicationic, which have a chemical structure formed by a hydrophobic tail, a spacer, and two positively charged heads (Figure 18a); and dianionic, whose heads have negative charges (Figure 18b).

### 5.2. Synthesis Pathways

Bicephalous surfactants have been synthetically obtained by the 1,4 addition Michael reaction, in step 1, using as precursorspropanolamine (1) and tert-butyl acrylate (2) in methanol under constant stirring at room temperature, followed by a series of steps (2–5) that are represented in the route of synthesis for this surfactant type, as reported by Kalhapure et al. (Figure 19) [146].

One year later, Ojewole et al. [149] made small modifications to the method proposed by Kalhapure. Initially, Ojewole et al. used the same propanolamine precursor with tert-butylacrylate under the same conditions and reaction medium (step 1). The difference consists of the use of other reactants for the reduction of the carboxylic acid group (step 2). They used hydrochloric acid (HCl), 4-dimethylaminopyridine (DMAP), methylene chloride (DCM), and 1-ethyl-3-(3-dimethylaminopropyl) carbodiimide (EDAC) (step 2). The next steps were similar under the same reaction conditions reported by Kalhapure et al. [147] (Figure 20).

More recently, Chaudhari et al. simplified the method of Kalhapure, using the same initial precursors and modifying the reaction media [150]. They achieved a reduction in the number of steps and reaction time (Figure 21).

On the other hand, Hanssan et al. proposed a three-step synthesis to obtain a quaternary bicephalous cationic surfactant through the Michael reaction, using a different precursor to the previously reported synthetic methods [151]. They used trihexylamine with tert-butylacrylate only using methanol (MeOH) (step 1). The product obtained reacts with triisopropylsilane (TIPs) and trifluoroacetic acid (TFA) (step 2), and finally the resulting product reacts with methyliodide (MeI) (step 3) to give as a result the cationic bicephalous surfactant (Figure 22).

A select group of researchers has reported the synthesis of bicephalous surfactants in a relative short period. Among the most notable groups working along this line, it can be mentioned the group of Kalhapure and Ojewole, who have revealed four novel surfactants of this type. For their part, Bazylinska et al. [152] have provided information on the synthesis of three new bicephalous surfactants. Figure 23 shows the chemical structures of some examples of bicephalous surfactants reported by these and other researchers.

### 5.3. Micelle Formation

The critical micellar concentration is an important parameter to characterize a surfactant since it describes the required concentration of this compound for the formation of stable micelles [146]. Two-headed surfactants, compared to conventional surfactants, have been shown to increase their surface activity by around a thousand times [148,150], due to their structure that has two hydrophilic heads and influences micellar formation, and at the same time requiring a lower concentration as compared to that of the equivalent monocephalous conventional surfactant to form micelles [146].

An example of the decrease in the CMC is the bicephalous dianionic surfactant called disodium (Z)-3,30-((3-(oleoyloxy) propyl) azanediyl) dipropanoate, which reduces the CMC by almost 50% compared to the conventional sodium oleate surfactant This corroborates that the CMC can be reduced by increasing the amount of polar heads in the chemical structure of the surfactant [146].

Micellar formation using a conventional surfactant is different from that resulting from a bicephalous surfactant. The main difference is that a greater amount of conventional surfactant is required to form a micelle (Figure 24a) than the bicephalous one (Figure 24b).

### 5.4. Applications

In 2013, Kalhapure et al. synthesized a bicephalous dianionic surfactant, which was used to prepare solid lipid nanoparticles with ketoconazole, a drug with low solubility and high permeability. They achieved a decrease in CMC (<50%) compared to a formulation prepared with sodium oleate, increasing the solubility, stability, biocompatibility, and biosafety of the drug [146,148]. 

For their part, Ojewole et al. obtained three bicephalous dianionic surfactants derived from oleic acid to obtain gels for oral administration of the antiretroviral drug didanosine. These surfactants were the 9-octadecenolic acid(9Z)-,3-[bis[3-(1,1-dimethylethoxy)-3-oxopropyl]amino] propyl ester, 9-octadecenolic acid(9Z)-,3-[bis(2-carboxyethyl) amino] propylester, and 9-Octadecenoic acid (9Z)-, 3-[bis(2-carboxyethyl) amino] propyl ester sodium salt. The results showed that the use of this type of surfactant increased the oral permeability of the drug compared to formulations where only oleic acid was used. It also was determined that an increase of the concentration of these surfactants caused an increase in oral drug permeation [149].

In another report, Bazylinska et al. used the bicephalous dianionic surfactant disodium N-dodecyliminodiacetate, C_12_N(COONa)_2_ to formulate normal microemulsions with isopropyl myristate and/or oleic acid as the oil phase, achieving a thermodynamically stable microemulsion. This microemulsion was shown to have zero toxicity in the in vitro study with gingival fibroblast (HGF) and skin keratinocyte (HaCaT) cell lines, proving its potential application in the cosmetic industry for treatments that require cutaneous administration [153]. Three years later, the same research group reported the use of this type of surfactant to encapsulate photosensitizers (meso-tetradenylporphyrin, TPP, and verteporphytin, VP). It was demonstrated that nanoencapsulation increased the solubility of the highly hydrophobic TPP and VP compounds, allowing their prolonged release and protection from photolytic degradation [124].

In 2015, Dhumal et al. developed a self-emulsifying drug delivery system with curcumin (an anticancer bioactive) using the (Z)-di-tert-butyl 3,3′-((3-112 (oleoyloxy)propyl)azanediyl)dipropanoate a bicephalous dianionic surfactant, which was obtained from oleic acid. They achieved the formation of a microemulsion that allowed to improve the solubility of curcumin at least 2.6 times when compared to a microemulsion obtained by using ethyl oleate. In addition, greater amounts of drug were loaded, and in vitro tests against the HeLa cell line showed that the formulation not only improved the solubility but also the permeability and bioavailability of curcumin [154].

Rambharose et al. used the bicephalous dianionic surfactant 9,12,15-octadecatrienoic acid, 3-[bis[3-(1,1-dimethylethoxy)-3-oxopropyl] amino] propyl ester, (9Z,12Z,15Z), whose precursor is a saturated fatty acid (linolenic acid), as an oil phase for the preparation of a nanoemulgel loaded with tenofovir, which is used in the treatment of human immunodeficiency virus. The permeation profiles were carried out and the results showed that the use of the bicephalous compound improved almost 40 times the permeation of the drug through the skin [155]. On the other hand, Chaudhari et al. obtained the bicephalous dianionic surfactant G0-PETIM dendron based on a bicephalous heterolipid (BHL), which was used for the preparation of microemulsions loaded with Efavirenz used for HIV treatment. The results indicated that using this surfactant, the solubility was improved 7.75 times compared to the use of only erucic acid oil. Furthermore, the release of the drug was increased at least six times, improving the bioavailability [150].

More recently, Hassan et al. synthesized ammonia-based quaternary bicephalic cationic surfactants to formulate quatsomes with a broad-spectrum antimicrobial drug (vancomycin) to be tested against methicillin-resistant *Staphylococcus aureus*. The toxic and hemolytic results demonstrated its biosafety and antimicrobial effect, which was improved up to eight times compared to a formulation without the use of the quaternary cationic bicephalic surfactant. In addition, sustained drug release was achieved by an intelligent system sensitive to pH 7.4 [151].

As can be seen above, there are no reports about the use of bicephalous surfactants in the polymer area, including synthesis and characterization; therefore, many areas of opportunity and research can be opened in this field in the future.

## 6. Conclusions

This review presents the synthesis, micelle formation, and applications status of two classes of non-conventional surfactants: gemini (dimeric) and bicephalous. The former is, by far, the most investigated of the two. Even though gemini surfactants have been known since 1935, commercial applications are scarce and conventional amphiphiles continue to dominate the market. Research has demonstrated that dimeric surfactants have improved properties over the conventional ones, which could lead to important savings in selected applications. The lower CMC indicates the potential to be advantageously applied, mainly in the cosmetic, pharmaceutical, oil recovery, catalysis, and polymer synthesis (emulsion polymerization) fields. Bicephalous surfactants represent a new generation of surface-active compounds that are starting to call to attention the need for future developments in applications dominated by conventional surfactants. Simplified synthetic routes for both gemini and bicephalous surfactants should be emphasized to open applications and markets in the near future.

## Figures and Tables

**Figure 1 ijms-23-01798-f001:**
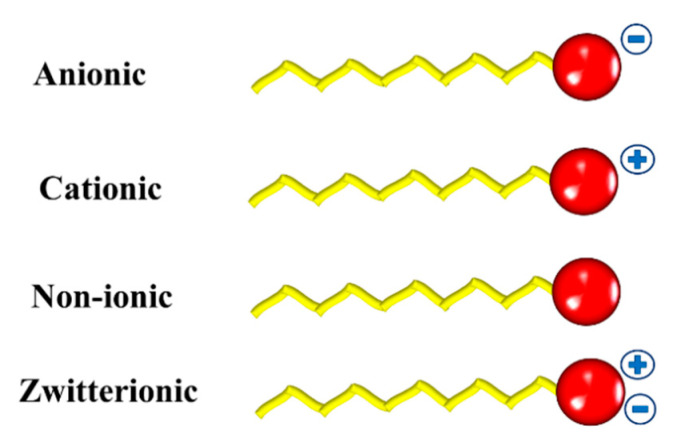
Schematic representation of the different types of surfactants.

**Figure 2 ijms-23-01798-f002:**
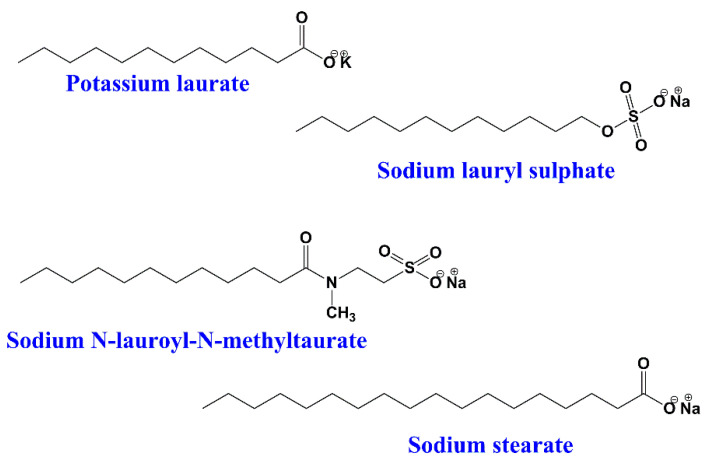
Chemical structures of some anionic surfactants.

**Figure 3 ijms-23-01798-f003:**
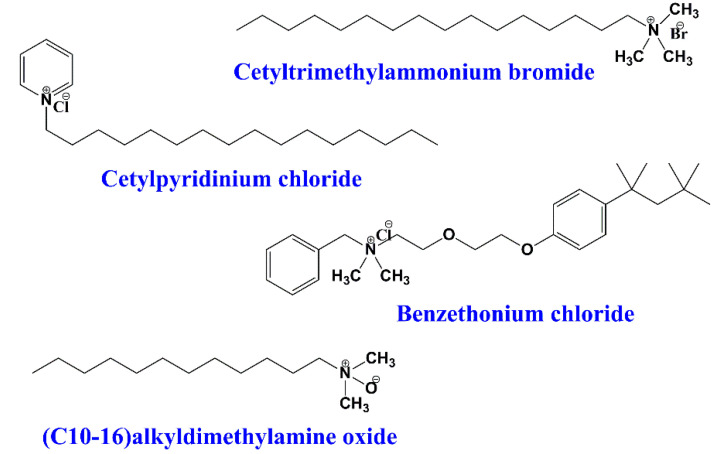
Examples of some common cationic surfactants.

**Figure 4 ijms-23-01798-f004:**
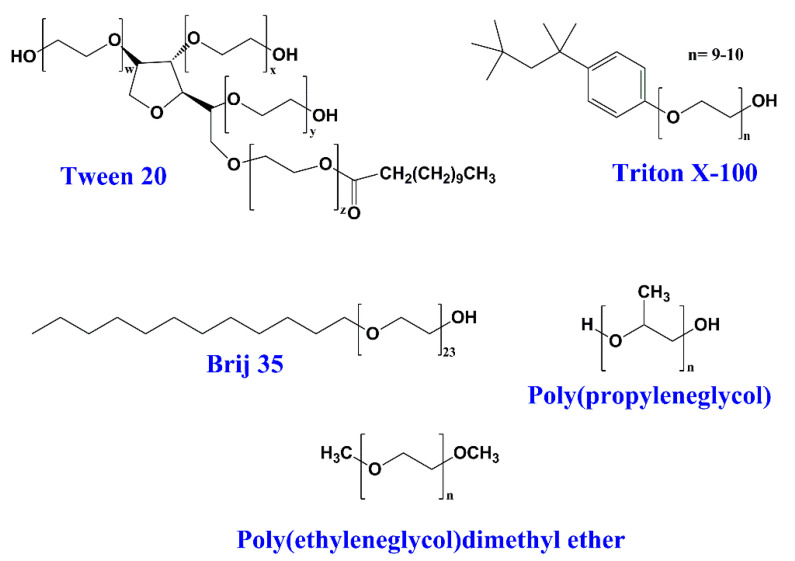
Chemical structures of common non-ionic surfactants.

**Figure 5 ijms-23-01798-f005:**
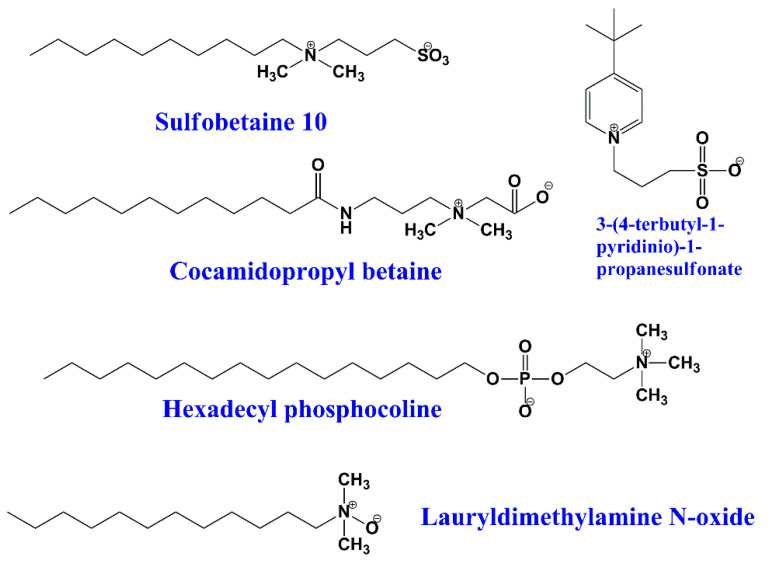
Chemical structures of some zwitterionic surfactants.

**Figure 6 ijms-23-01798-f006:**
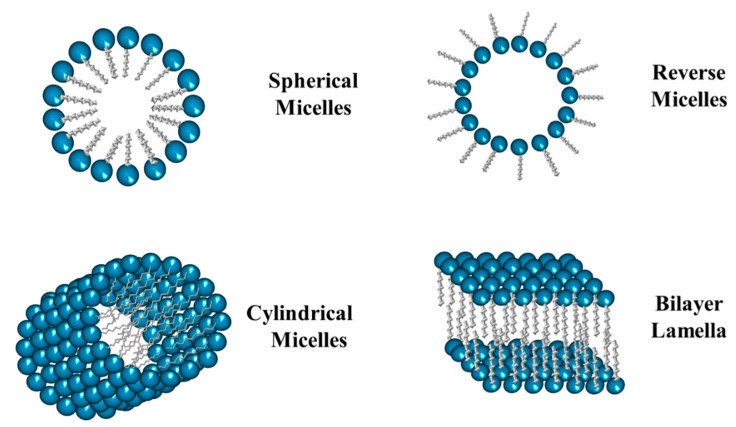
Schematic representation of different types of micelles.

**Figure 7 ijms-23-01798-f007:**
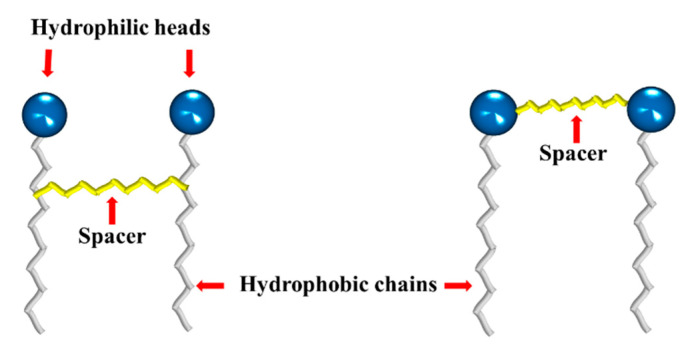
Typical structures of gemini surfactants.

**Figure 8 ijms-23-01798-f008:**
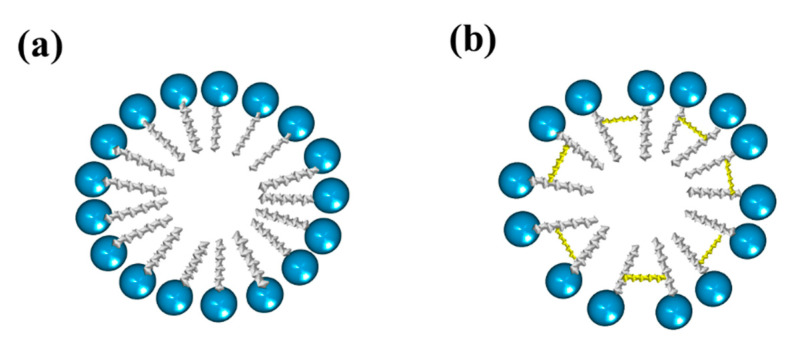
Representation of micelles formed from (**a**) conventional surfactants and (**b**) gemini surfactants.

**Figure 9 ijms-23-01798-f009:**
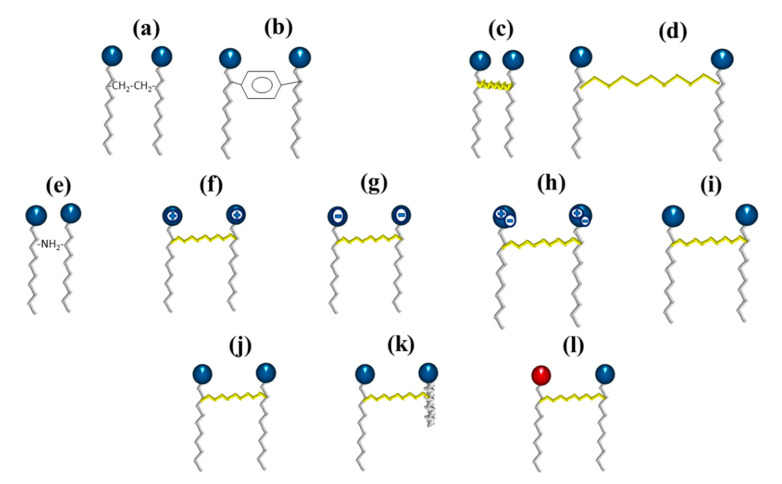
Schematic representation of the different types of gemini surfactants: (**a**) flexible spacer, (**b**) rigid spacer, (**c**) short chain spacer, (**d**) long-chain spacer, (**e**) polar spacer, (**f**) cationic, (**g**) anionic, (**h**) zwitterionic, (**i**) non-ionic, (**j**) two identical hydrophilic heads and hydrophobic chains, (**k**) two non-identical hydrophobic chains, and (**l**) two non-identical hydrophilic heads.

**Figure 10 ijms-23-01798-f010:**
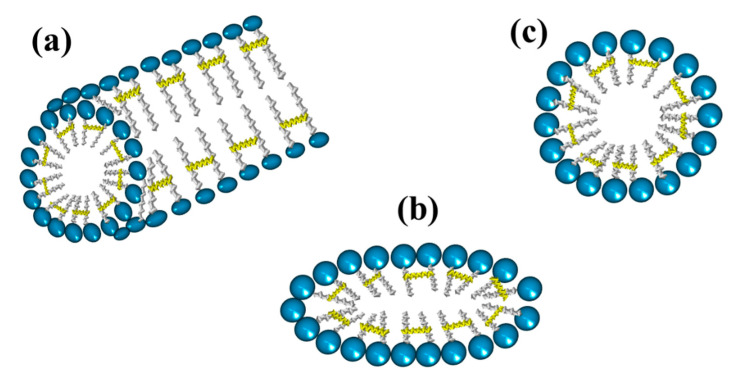
Geometries of micelles from gemini surfactants linked by the tail: (**a**) fiber-like, (**b**) elliptical, and (**c**) spherical.

**Figure 11 ijms-23-01798-f011:**
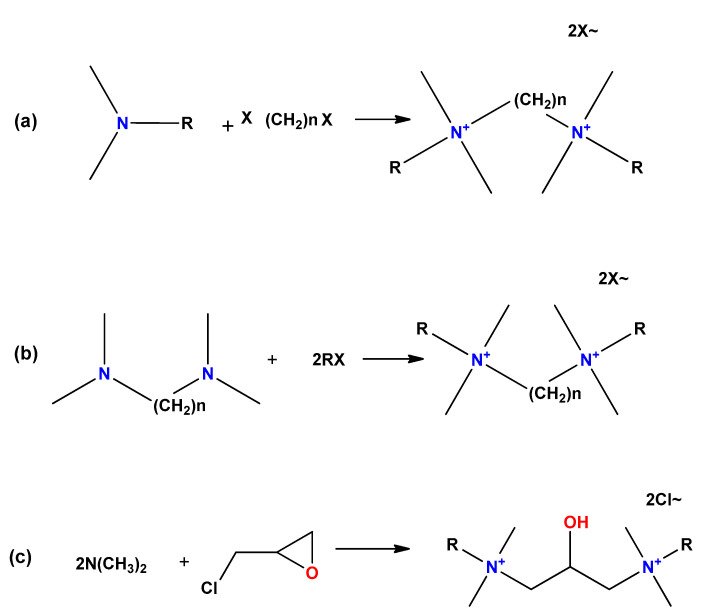
General routes to obtain gemini surfactants. (**a**) Reaction of long chain tertiary amines with dihalogenated substrates as organic di-bromides or dichlorides; (**b**) reaction of N,N,N′,N′-tetramethylpolymethylene diamines with alkyl halides; and (**c**) reaction of long chain tertiary amines with a haloalkylene oxide substrate.

**Figure 12 ijms-23-01798-f012:**
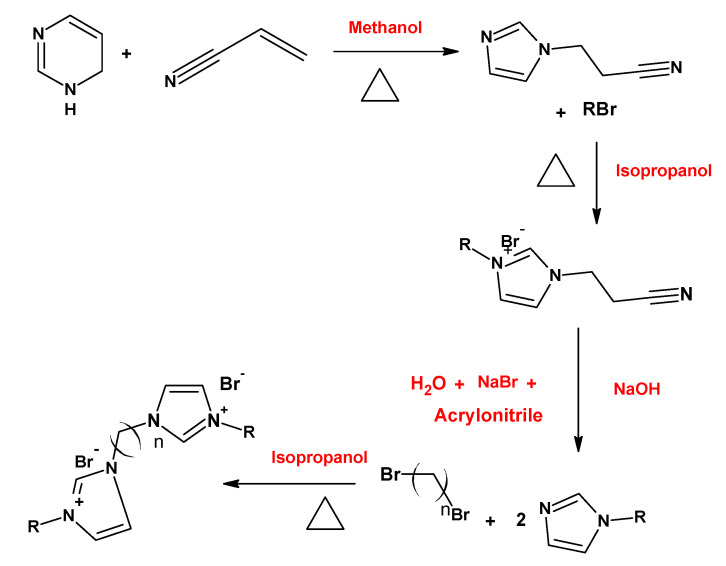
Route of synthesis of the imidazolium gemini surfactants synthesized by Ren et al. [85].

**Figure 13 ijms-23-01798-f013:**
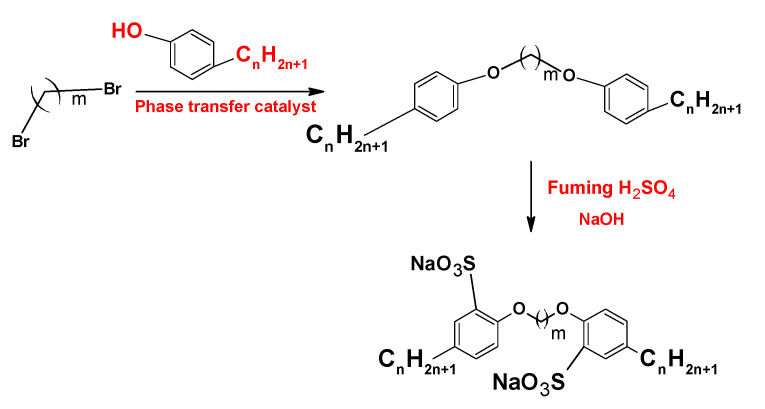
Synthesis of sulfonate gemini surfactants.

**Figure 14 ijms-23-01798-f014:**
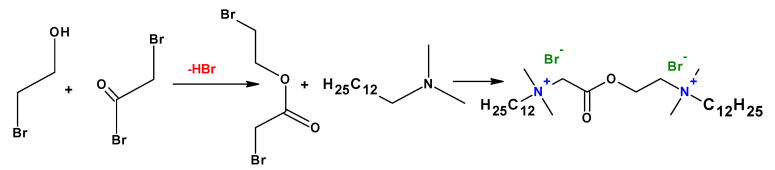
Synthesis of gemini surfactants containing an ester group.

**Figure 15 ijms-23-01798-f015:**
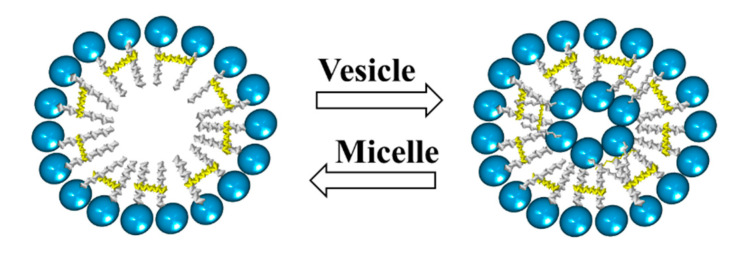
Scheme of transition from micelles to vesicles and vice versa.

**Figure 16 ijms-23-01798-f016:**
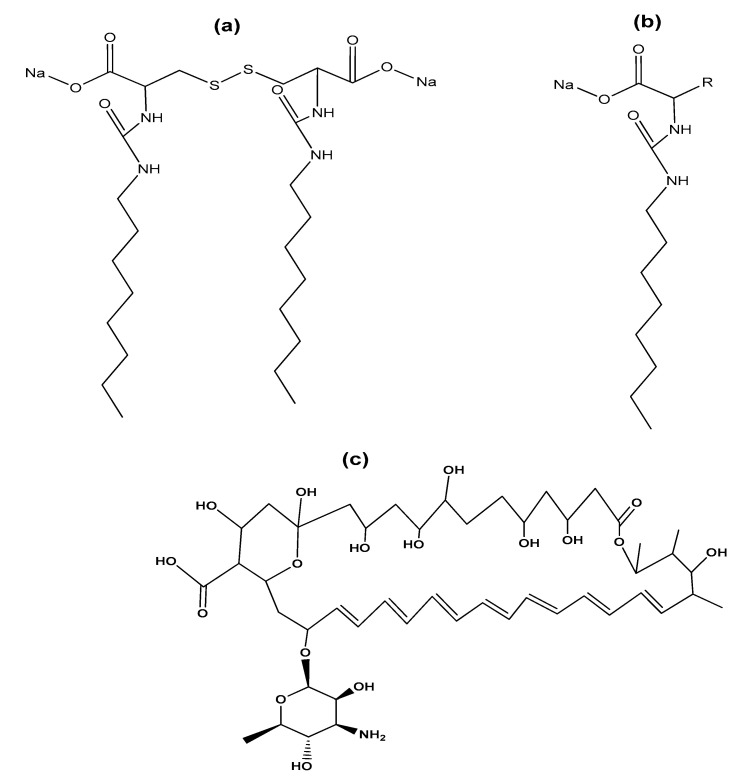
(**a**) Anionic gemini surfactant derived from cysteine and (**b**) its monomeric counterpart; (**c**) chemical structure of antifungal polyene antibiotic amphotericin B.

**Figure 17 ijms-23-01798-f017:**
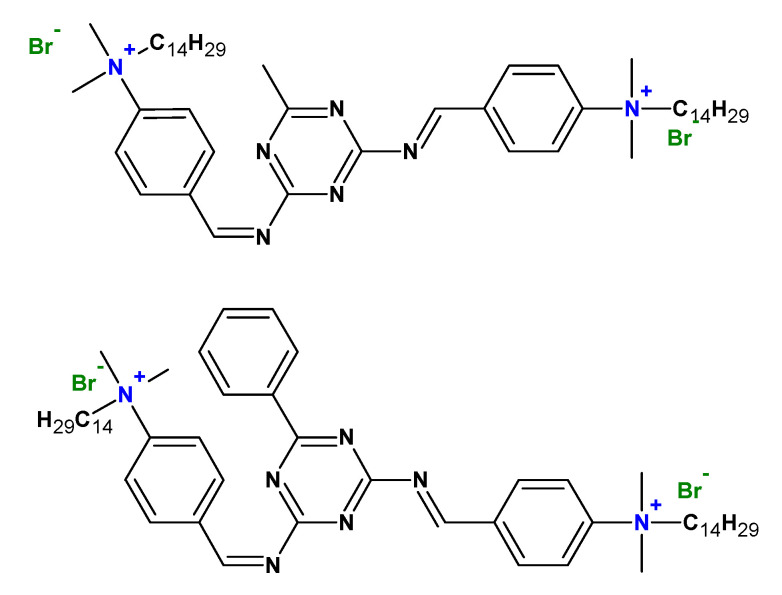
Chemical structures of some gemini surfactants used as corrosion inhibitors.

**Figure 18 ijms-23-01798-f018:**
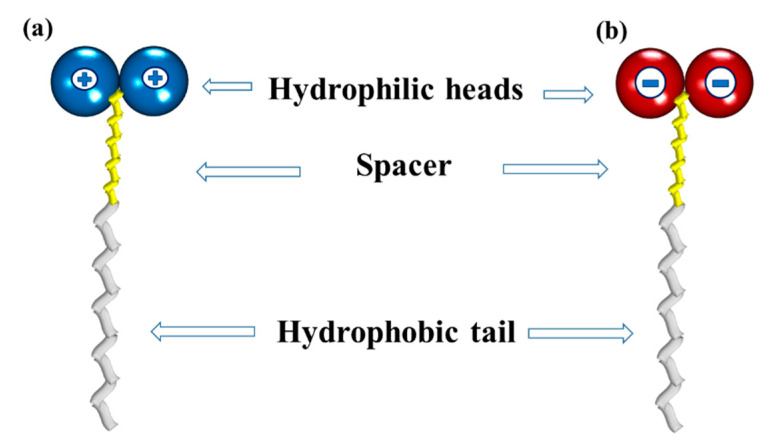
Schematic representations of (**a**) a bicephalous dicationic surfactant; and (**b**) a bicephalous dianionic surfactant.

**Figure 19 ijms-23-01798-f019:**
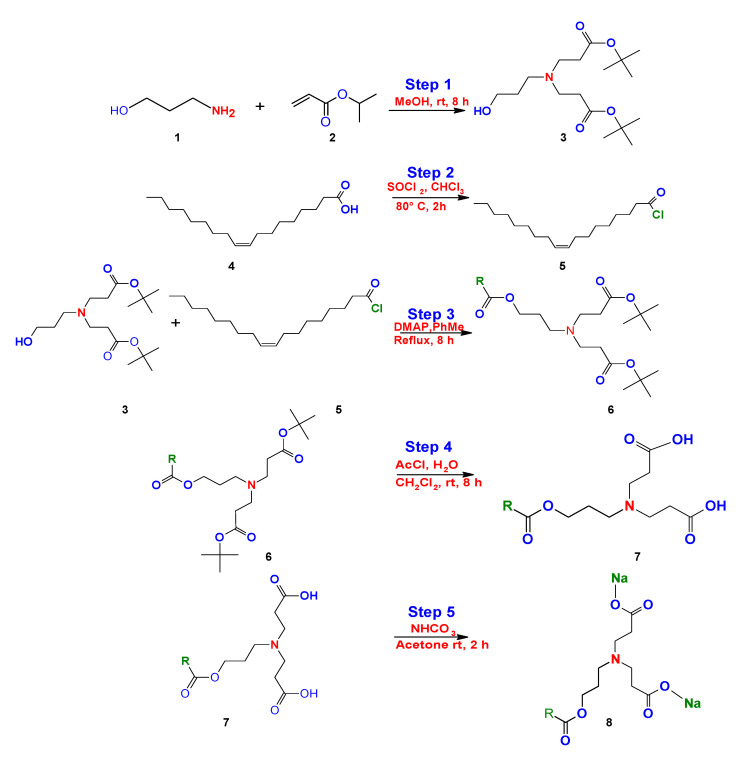
Synthesis of bicephalous dianionic surfactant proposed by Kalhapure et al. [147].

**Figure 20 ijms-23-01798-f020:**
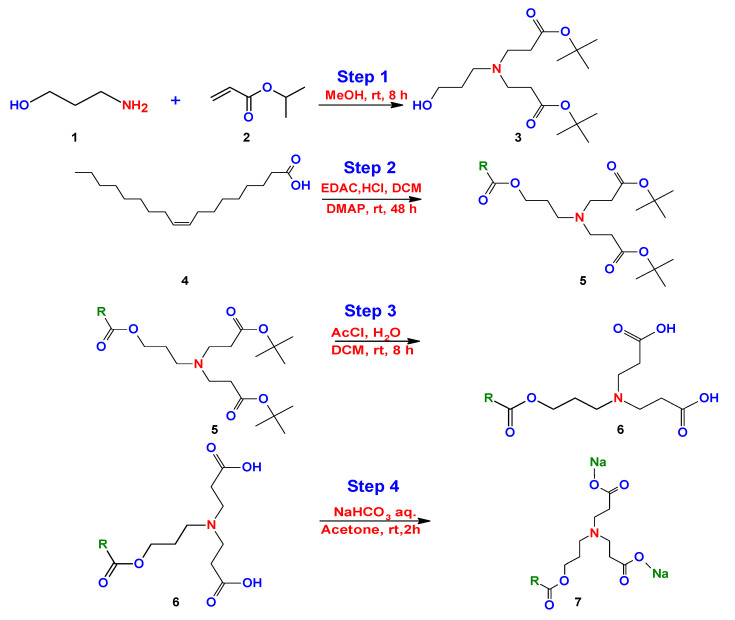
Synthesis of bicephalous dianionic surfactant proposed by Ojewole et al. [148].

**Figure 21 ijms-23-01798-f021:**
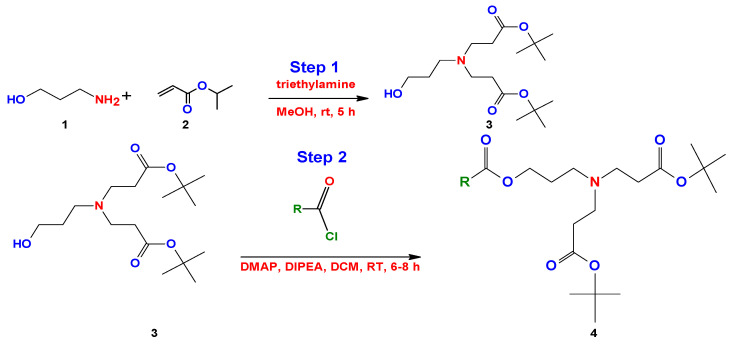
Synthesis of bicephalous dianionic surfactants proposed by Chaudhari et al. [150].

**Figure 22 ijms-23-01798-f022:**
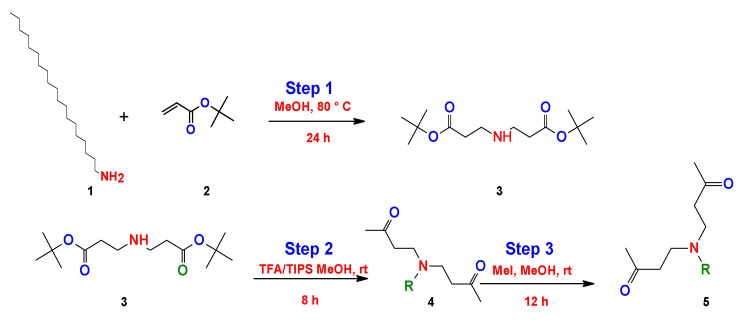
Synthesis of the bicephalous cationic surfactants proposed by Hassan et al. [151].

**Figure 23 ijms-23-01798-f023:**
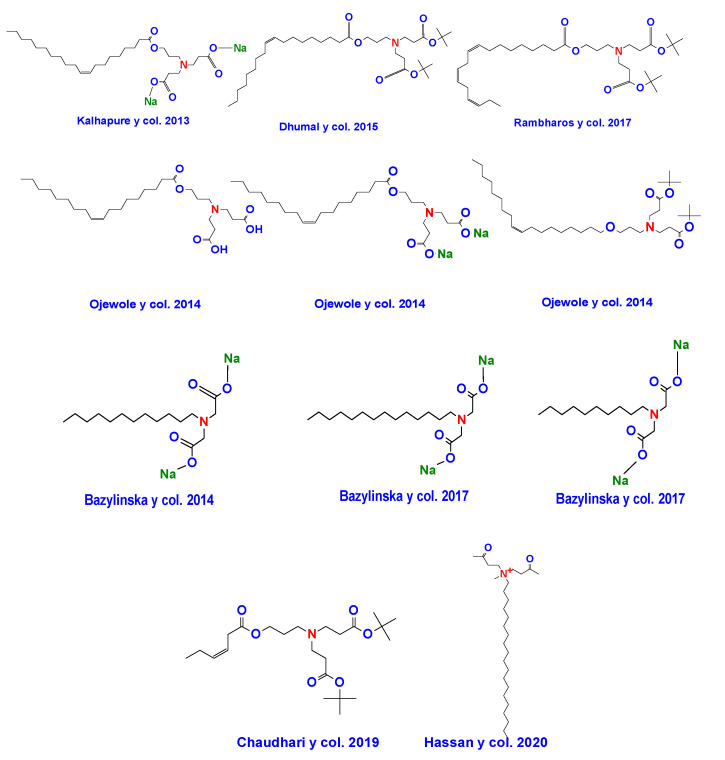
Chemical structures of some bicephalous surfactants reported in recent years.

**Figure 24 ijms-23-01798-f024:**
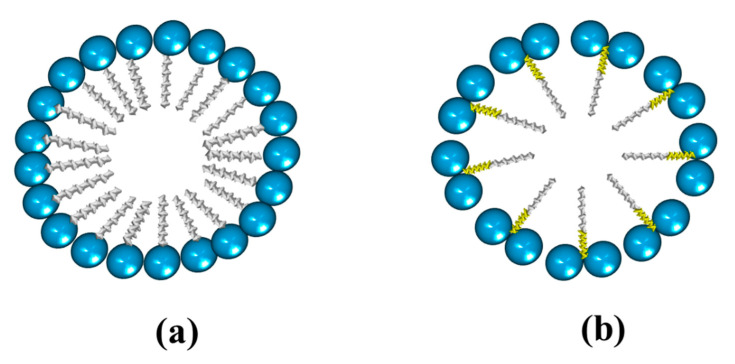
Micelle structure from (**a**) a conventional surfactant; and (**b**) a bicephalous surfactant.

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
