# Peer review of "Gemini and Bicephalous Surfactants: A Review on Their Synthesis, Micelle Formation, and Uses"

_ijms, 2022, doi:10.3390/ijms23031798_

Round 1

Reviewer 1 Report

  1. Abstract should be elaborated and brief description of the applications of the gemini as well as bicephalous surfactants should be given.
  2. Lines 35&36: sentence should be rewritten and the both terms i.e., surface tension (at air water interface) and interfacial tension (at the interface between two liquids) should be described separately.
  3. Figure 3: write the complete name of CTAB.
  4. In the properties of micelles, there is no discussion regarding the catalytic properties of the micelles. Micellar catalysis is an important aspect of the micelle properties and this part, particularly related to the gemini surfactants, must be added in the review. 
  5. Furthermore, gemini surfactants show interesting properties when interact with proteins in comparison to their conventional counterparts and authors are advised to include a brief account of the comparison of their interaction with proteins. 
  6. A through language editing is also required. 

Author Response

We are very grateful to the reviewers for their thoughtful and helpful comments. We have addressed their comments as detailed below. Changes in the revised manuscript (ijms-1475278) are in red color.

REVIEWER 1

Comments and Suggestions for Authors

Abstract should be elaborated and brief description of the applications of the gemini as well as bicephalous surfactants should be given.

Answer: Thanks for your suggestion. Applications of these surfactants have been mentioned in the abstract.

Lines 35&36: sentence should be rewritten and the both terms i.e., surface tension (at air water interface) and interfacial tension (at the interface between two liquids) should be described separately.

 Answer: Thanks for your comment. This sentence has been changed.

Figure 3: write the complete name of CTAB.

Answer: It has been done

In the properties of micelles, there is no discussion regarding the catalytic properties of the micelles. Micellar catalysis is an important aspect of the micelle properties and this part, particularly related to the gemini surfactants, must be added in the review. 

Answer: We very much appreciate reviewer’s suggestion and we have added this particular point in section 4.6 of the review as well as some references.

Furthermore, gemini surfactants show interesting properties when interact with proteins in comparison to their conventional counterparts and authors are advised to include a brief account of the comparison of their interaction with proteins. 

Answer: We have added this point and some references in the revised version (see section 4.6).

 A through language editing is also required. 

Answer: Manuscript has been thoroughly revised in order to improve the language.

Reviewer 2 Report

Gemini and Bicephalous Surfactants: A Review on Their Synthesis, Micelle Formation, and Uses (ijms-145278)

Int. J. Molecular Sciences

This is a conventional revision about different type of surfactants, their syntheses and their applications. A revision about the formation of micelles in also included.

From my point of view, the most interesting section of the manuscript begins when the authors show the gemini and bicephalous surfactants. The latter are the most recent surfactants.

Perhaps the discussion of the manuscript could be improved. In any case, the manuscript could be accepted for publication in the actual version.

Author Response

We are very grateful to the reviewers for their thoughtful and helpful comments. We have addressed their comments as detailed below. Changes in the revised manuscript (ijms-1475278) are in red color.

REVIEWER 2

Comments and Suggestions for Authors

Gemini and Bicephalous Surfactants: A Review on Their Synthesis, Micelle Formation, and Uses (ijms-145278)

Int. J. Molecular Sciences

This is a conventional revision about different type of surfactants, their syntheses and their applications. A revision about the formation of micelles in also included. From my point of view, the most interesting section of the manuscript begins when the authors show the gemini and bicephalous surfactants. The latter are the most recent surfactants.

Perhaps the discussion of the manuscript could be improved. In any case, the manuscript could be accepted for publication in the actual version.

Answer: Thanks for your comments. We have modified the discussion.

Round 2

Reviewer 1 Report

The paper is  now publishable in IJMS .